# Learning to be Multimodal : Co-evolving Sensory Modalities and Sensor Properties

**Rika Antonova**
Stanford University
`rika.antonova@stanford.edu`

**Jeannette Bohg**
Stanford University
`bohg@stanford.edu`

**Abstract:** Making a single sensory modality precise and robust enough to get human-level performance and autonomy could be very expensive or intractable. Fusing information from multiple sensory modalities is promising – for example, recent works showed benefits from combining vision with haptic sensors or with audio data. Learning-based methods facilitate faster progress in this field by removing the need for manual feature engineering. However, the sensor properties and the choice of sensory modalities is still usually done manually. Our blue-sky view is that we could simulate/emulate sensors with various properties, then infer which properties and combinations of sensors yield the best learning outcomes. This view would incentivize the development of novel, affordable sensors that can make a noticeable impact on the performance, robustness and ease of training classifiers, models and policies for robotics. This would motivate making hardware that provides signals complementary to the existing ones. As a result: we can significantly expand the realm of applicability of the learning-based approaches.

**Keywords:** co-evolution, multimodal sensing

## 1 Motivation and Overview

Animals have evolved an impressive variety of sensors. A few examples: vision in the visible light spectrum and beyond (snakes seeing in infrared, bumblebees and reindeer in UV); sound emitters/receptors for ranges beyond the audible (elephant low-frequency calls to reach mates miles away; high-frequency echo-location in bats), electroreception for underwater localization (in dolphins, sharks, platypus); sensing Earth's magnetic field for navigation (in pigeons, turtles, salmon). If 'more-is-better' were the main law of sensing, then we would expect to see impressive agglomerations of sensory capabilities in the more advanced animals. Instead, sensors often seem specialized to the class of tasks an animal needs to perform to survive. Thus, sensors likely co-evolved with the organisms and, in part, defined the class of tasks that each organism could accomplish.

To build sensors for robotics we usually do task-agnostic optimization: maximize speed, accuracy and robustness while minimizing size, cost and energy consumption. With the recent success of machine learning, we now have scalable neural networks (NNs) that can learn to construct representations from raw sensor data. This can drastically improve both speed and precision/success rate. For example, NNs can be trained to work on partial and noisy point clouds [1, 2, 3, 4], eliminating the need for shape completion. Shape completion is computationally expensive and can lead to excessive thickening, which in turn limits the potential of active learning approaches (lack of perception speed) and can degrade manipulation with thin objects (lack of point cloud precision).

A simplistic view is that we can include all the available sensory data and rely on end-to-end learning. However, such view could significantly impede the long-term progress. Below we highlight two major issues and propose how to address them by employing a co-evolution approach to sensing.

**The need for smart compromises**: One major issue is that we cannot literally include in our hardware platforms all possible sensors: this is intractable at least due to cost and size. Hence, we need to make compromises: select a subset of sensors given the budget, size and energy constraints. The choice of which sensors to buy/include is currently done manually. This could have been sufficient in the past: the number of available sensors was small; the scope of the robotics tasks was limited

Blue Sky Papers, 5th Conference on Robot Learning (CoRL 2021), London, UK.

(structured tasks in isolated environments); traditional approaches used sensor data in fixed ways. For example, for a specific task and known objects one could select a fingertip force sensor that would report normal force that an object exerts on the gripper's finger. This 1D signal could be enough to guide the manipulation if the initial pose of the object was known. In contrast, many robotics researchers now strive to handle objects on unknown shape, material and weight in unstructured environments, striving to achieve this by automatically learning NN-based representations from high-dimensional sensor data, such as RGBD camera images. The downside of this generality is that the learned representations are not comprehensible at a glance. Hence, it is difficult to manually identify sensory modalities and properties that would be necessary and sufficient for a certain class of tasks. As a result, we usually end up including sensors that happen to be available, rather than searching for sensors that would be optimal for the class of tasks we aim to solve.

**The need for smart sensing**: Suppose we make a bet on sensors becoming smaller and cheaper, hoping to include more and more sensory modalities in the future. Can we then expect that end-to-end learning-from-scratch would suffice? Many academic researchers do not have access to large-scale compute resources, hence their research would be hampered if large-scale training is required in all cases. Generality and flexibility are not always enough for those working with robot hardware and limited computational resources. Hence the need to retain only the sensory data crucial for supporting a given research agenda, and focus compute resources on the relevant signals, without worrying that a key modality is omitted. Smart hardware design could turn even a low-dimensional combination of signals into a highly efficient hybrid sensory modality. For example, in experiments with the Exploratory Hand [5], adding basic tactile sensing greatly improved contact and object identification when combined with contact particle filtering. Roboticists can deduce what kind of sensing could be useful from model-based (physics) considerations. However, they can only test a few target settings/tasks manually. A more streamline way would help to quickly gauge which novel components are effective, which combinations have the potential to produce low-cost and accurate results.

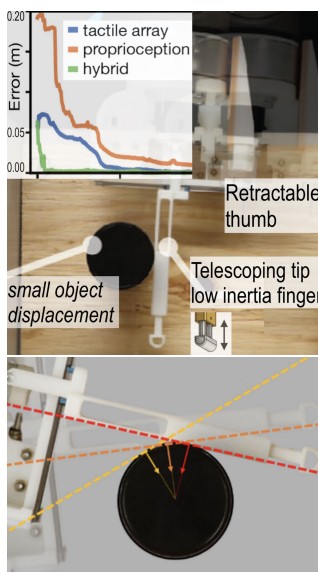

Figure 1: Exploratory hand [5]

## 1.1   Co-evolving sensors to optimize efficiency of learning algorithms that use sensor data

Our blue-sky paradigm is to integrate development of sensor hardware (and low-level signal processing) with research on using various sensory modalities for learning-based approaches. We can start by creating approximate models of various sensors in simulation, then feed the synthetic sensor data to the target learning methods, and attempt to train on tasks that represent a class of problems of interest. We could vary sensor properties, such as speed and precision, to determine which aspects have significant effects on the training performance. With this, we could infer the minimum requirements for the sensors and find combinations of modalities that enable successful learning. There has been progress in sensor modeling [6, 7, 8], but relying only on simulation is limiting, since it remains challenging to fully close the sim-to-real gap [9]. Hence, after initial simulation-based analysis, it would be crucial to use hardware data. We could collect sensor data from expensive sensors (e.g. high-speed cameras, high-resolution tactile arrays) and identify the maximum level or degradation that still allows learning useful representations. We could also search for optimal combination of sensors: employ sensitivity analysis to determine modalities that contribute the most to the learning success, use explainability methods to show contributions of different sensory modalities.

To motivate the above paradigm, let us consider one concrete challenging example: manipulation of highly deformable objects. More than a decade ago, it has been shown that high-speed cameras and high-speed actuation can help with higly dynamic tasks [10, 11]. However, such systems did not gain widespread use, either due to their high cost or safety concerns of high-speed actuation. We could search for alternative combinations. Perhaps high-speed actuation could be replaced by learning multistep dynamics models: anticipate the motion of the object instead of reacting instantly to its current state. Perhaps such multistep feedforward models would be an inexpensive replacement for the feedback-based approaches that need to rely on high-speed sensing and actuation. We could also test whether feedback-based approaches relying on high-speed cameras can cope with occlusions. If not, we could search for complementary modalities, such as tactile sensing. We could analyze

whether adding fingertip sensing is enough. It may turn out that, for some classes of tasks, it is essential to add sensing to other parts of the robot hands/grippers/arms/body, suggesting the use of sensors closer to an inexpensive 'skin'. If using robotic skin turns out to be highly beneficial, we could identify the minimum requirements for such sensors for a given set of tasks. This paradigm would enable finding an optimal combination of the properties of a vision system (speed, number of cameras) and tactile system (resolution, placement). It would allow to jointly minimize the sensor cost and size requirements, while still ensuring that the learning algorithms get a signal that is sufficiently rich for accomplishing a set of target tasks. We might discover that the well-performing combinations require sensors that have not been built yet. For example, we might find that reporting full contact information with all parts of the grippers in simulation is highly beneficial for successful training, hence demonstrating the potential benefits of developing a sensitive skin that can cover a large part of the grippers. By showing that learning succeeds when using such sensors we could help the teams working on sensor hardware to obtain additional funding and support for the development.

## 2 Acquiring Novel Sensors and Sensory Modalities

**High-speed / high-coverage sensors**: [10] showed impressive results on dynamic tasks by combining fast actuation (180 degrees per 0.1 second) and high-speed cameras with 500 frames per second (FPS) rate to solve advanced tasks, such as in-the-air cloth folding and knot tying. High-speed tactile sensors (1kHz) enabled dynamic in-hand manipulation tasks, such as pen spinning [12]. LIDAR sensors have been tested at high speeds for race driving ($>150$ km/h), showing potential for high-speed localization [13]. High-speed vision sensors is a recurring interest in robotics and computer vision.

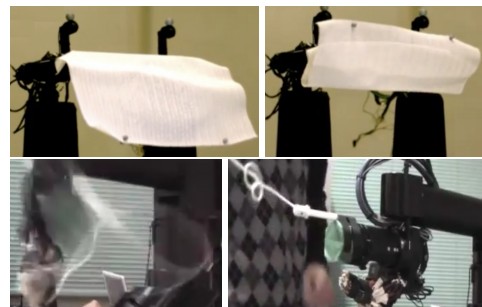

Figure 2: In-the-air cloth folding, knotting [10, 11]

Systems beyond 500 FPS are expensive, moreover, caching and transmitting at high rates can be problematic [16]. One solution is to compress the signal at the time of sensing [17]. Another solution is event-based methods: report only the salient changes [18, 19, 20]. This could allow high-resolution sensing without overwhelming the learning methods with redundant data. This could be particularly beneficial for high-coverage skin sensors, since we know that humans rely on such ability (e.g. event-based aspects of fast-adapting mechanoreceptors on human fingertips [21]; sensing clothing only when we initially put it on). If we optimize such functionality separately from understanding how the signals are used by the learning methods – we risk creating input data that is difficult to interpret. Co-evolving sensor properties together with the learning methods would help ensure the overall training success.

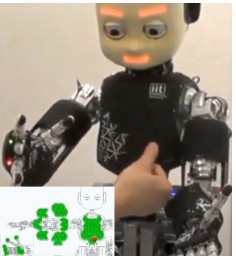

Figure 3: ≈ 2000 skin sensors on iCub [14, 15]

**More than the sum of its parts**: A number of recent works showed that combining multiple sensory modalities is beneficial. Vision (from conventional 30Hz cameras) has been combined with tactile and haptic sensors [22, 23, 24, 25, 26], range sensors [27, 28], auditory data [29, 30, 31, 32, 33]. Withholding certain modalities can be detrimental even for the most powerful and adaptive learning systems – the human brain. Research in prosthetics shows that the absence of tactile information can lower device adoption rates [34]; incorporating tactile sensing is important for making it easy for the brain to adapt to using the hand and arm prosthetics [35, 36]. As the available learning algorithms evolve, it is important to keep track of which modalities are necessary and sufficient for the new state-of-the-art learning algorithms to succeed. Following the co-evolution paradigm would allow us to jointly optimize the individual sensor properties and experiment with various combinations of sensory modalities. As a result, we could discover the need for new hybrid sensory modalities. For example, if high-speed visuo-tactile sensors show promise, but face issues with caching, transmission and synchronization – we could argue for developing an integrated sensing system. Placing vision and tactile sensing on the same platform/chip would reduce cost, communication overhead, improve synchronization, allow integrated compression and event-driven transmission (FingerVision [37] showed some of these benefits). The co-evolution paradigm would let us present a convincing case by clearly showing the expected benefits on the learning system. This could strengthen the motivation and enthusiasm for funding the hardware development. Once affordable hardware

is available, the learning-for-robotics approaches would be able to show the benefits in hardware experiments, thus furthering the adoption of advanced learning-based methods in core robotics.

**The 6th sense and beyond**: It can be easy to settle into a local optimum of using a common inexpensive set of visual and tactile sensors. To overcome this, the co-evolution view can help perform initial estimation of what the novel sensing modalities could offer. Consider the recently developed techniques of 'seeing through walls' with WiFi [38] and radio signals [39]. We could make a rough model of these by post-processing real fully-observable data to approximate the quality of the data that such sensors would output under occlusions. We can then quickly find out whether such novel sensors can significantly improve the performance of the learning approaches. Similarly, we can explore other unconventional sensors, for example, the miniature versions of spectrometers [40, 41]. Finally, we can compare the benefits of multistep predictive models versus using high-speed hardware sensors. If multistep models can anticipate the near future well enough to replace a high-speed sensor – then we can spend resources on acquiring data and compute power to train such models.

## 3 Optimization for the Co-evolution of Sensors

**Global search for optimal sensing**: To avoid the difficulties of modeling the physics of sensors, we can instead create simulated data streams that match the precision, speed and basic noise patterns of existing or hypothetical/new sensors. To find combinations of sensory modalities and sensor properties that yield the best performance, we can use data-efficient global search methods, e.g. a recent variant of Bayesian optimization (BO) for mixed categorical-continuous spaces [42]. We can start by sampling a batch of $N$ sensor combinations $\{s_1^{(n)}, ..., s_C^{(n)}\}_{n=1}^N$ and properties $\{\boldsymbol{x}_1^{(n)}, ..., \boldsymbol{x}_C^{(n)}\}_{n=1}^N$. Here, each $s_c$ is a categorical variable indicating sensor type, and $\boldsymbol{x}_c$ is a vector of sensor characteristics (sampling rate, noise level, resolution, etc). We can launch $N$ training runs in parallel on a set of target tasks, then update BO posterior with results. This posterior would model the objective function $f(combo)$, expressing the expected performance of the learning algorithms that use a given combination of sensors. With BO we also get uncertainty estimates. If the posterior shows benefits from a novel combination of sensors, we can run optimization further until the uncertainty shrinks enough to present a clear case for purchasing the new sensors or developing the new hardware. More generally, we can view the search for optimal combination as hierarchical hyperparameter optimization and apply a variety of relevant methods [43, 44]. Multi-objective optimization [45] could find a Pareto frontier for optimizing performance versus cost to accommodate various budget levels.

**Incorporating real data**: To avoid over-reliance on simulation, we need to use real sensor data when possible. For example, we could construct a dataset that pertains to the aspects of interest/tasks from a high-resolution/speed sensor. Data collection could be done in a lab that develops the high-resolution/speed sensors. Other labs can use the dataset to mimic lower-resolution/speed (by degrading the sensor readings) to find the sensor resolution/speed that is sufficient for learning.

## 4 Relevant Work in Morphological Computation and Explainability

**Co-evolving morphology**: Recent works in searching for optimal morphology provide a level of assurance that our proposed co-evolution approach would be tractable in practice. [46] provide a way to evolve morphology of a rigid hand; [47, 48] develop automated co-design of soft hand morphology and grasping controllers; [49] propose a way to specialize sensor readings for a soft hand for new tasks. While these works do not suggest a longer-term strategy for sensor evolution, they do show that co-evolving hardware and algorithms (e.g. controllers) is beneficial and tractable. The classical sensor selection problem [50, 51] is related, though it views sensor properties as fixed.

**Explainability**: To analyze the contributions of various sensor modalities we can leverage recent work in explainability for deep learning [53, 54, 55, 56, 57]. This would be especially useful for sensors that appear promising, but have not been widely adopted yet. Visualizing the aspects of sensor readings that make a large impact on the learned models would allow sensor developers to focus on enhancing these particular aspects.

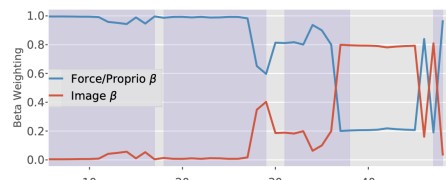

Figure 4: Learning when to use force readings (blue) on contact (shaded) vs images (red) [52]

**Acknowledgments**

Toyota Research Institute provided funds to support this work. Rika Antonova is supported by the National Science Foundation grant No.2030859 to the Computing Research Association for the CIFellows Project.

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
