# OpenReview forum: "Learning to be Multimodal : Co-evolving Sensory Modalities and Sensor Properties"
_robot-learning.org/CoRL/2021/Conference/Blue_Sky — CoRL 2021, Blue Sky_

### Official Review · Reviewer_hn4u · 2021-08-29

**Novelty:** Good
**Impact:** 3
**Clarity Of Presentation:** Good

**Recommendation:**

Weak Accept: I recommend accepting the paper, but will not argue for my recommendation if the majority of other reviewers have a different opinion.

**Summary:**

This paper aims to address the issue of how to improve the performance for a robot perceving its surroundings, with the emphasizing on maximally exploiting complementary properties of equipped sensors of a robot, under a robot learning style instead of commonly used mannual methods.

**Summary Of Recommendation:**

The issue this paper addressed is really interesting, the main idea of this paper is clearly described. I also believe learning-based approaches might be the right way to this issue, since traditional manual method do requires very rich expert experiences, which is costly and sometimes intractable.

---

### Official Review · Program_Chairs · 2021-09-29

**Novelty:** Good
**Impact:** 2
**Clarity Of Presentation:** Very Good

**Recommendation:**

Weak Reject: I recommend rejecting the paper, but will not argue for my recommendation if the majority of other reviewers have a different opinion.

**Summary:**

This paper argues for the automated design of multiple sensor modalities in the context of a learning task, thus making the sensor design task-aware. It then phrases the design problem as that of optimization over the design space. This general idea is known, at least, when executed manually.  However, this paper  then discusses its potential  on deformable object manipulation.

**Summary Of Recommendation:**

Overall, joint or iterative optimization of sensor modalities and the learning method presents an interesting possibility. It would be good to identify specific robot tasks where this joint optimization can make a critical difference. Also the global search in the sensor design space is expensive. Every step requires (potentially) reconfiguring/rebuilding the sensor hardware. What is in the current manuscript is a possibility in principle, but lacks any indication of feasibility.

---

### Decision · Program_Chairs · 2021-10-01

**Decision:**

Accept

**Comment:**

Both reviewers appreciated the high level idea presented in the paper and found it very interesting. The paper is a good fit for the Blue sky track.